# Phytochemical Screening and Evaluation of Antioxidant Properties and Antimicrobial Activity against *Xanthomonas axonopodis* of *Euphorbia tirucalli* Extracts in Binh Thuan Province, Vietnam

**DOI:** 10.3390/molecules26040941

**Published:** 2021-02-10

**Authors:** Nguyen Thị My Le, Dang Xuan Cuong, Pham Van Thinh, Truong Ngoc Minh, Tran Dinh Manh, Thuc-Huy Duong, Tran Thi Le Minh, Vo Thi Thu Oanh

**Affiliations:** 1Faculty of Fisheries, Ho Chi Minh University of Food Industry, 140 Le Trong Tan, Tan Phu District, Ho Chi Minh 70000, Vietnam; 2Faculty of Biological Sciences, Nong Lam University, Ho Chi Minh City, Thu Duc 71300, Ho Chi Minh 70000, Vietnam; 3Nha Trang Institute of Technology Application and Research, Vietnam Academy of Science and Technology, Nha Trang 650000, Vietnam; biofoodchemtech@gmail.com; 4Faculty of Biology, Graduate University of Science and Technology, Vietnam Academy of Science and Technology, Ha Noi 100000, Vietnam; 5Faculty of Food Science and Technology, Ho Chi Minh University of Food Industry, 140 Le Trong Tan, Tan Phu District, Ho Chi Minh 70000, Vietnam; thinhpv@hufi.edu.vn; 6Center for Research and Technology Transfer (CRETECH), Vietnam Academy of Science and Technology, Cau Giay District, Hanoi 100000, Vietnam; minhtn689@gmail.com; 7Institute of Applied Mechanics and Informatics, Vietnam Academy of Science and Technology, Cau Giay District, Hanoi 100000, Vietnam; 8Faculty of Food Technology, Thu Dau Mot University, Thu Dau Mot City, Binh Duong 820000, Vietnam; manhtd@tdmu.edu.vn; 9Deparment of Chemistry, Ho Chi Minh University of Education, District 5, Ho Chi Minh 77000, Vietnam; huydt@hcmue.edu.vn; 10Faculty of Agronomy, Nong Lam University, Ho Chi Minh City, Thu Duc 71300, Ho Chi Minh 70000, Vietnam; vtthuoanh@hcmuaf.edu.vn

**Keywords:** *X. axonopodis*, *E. tirucalli*, phytochemical, ethyl acetate, antibacterial activity

## Abstract

*Euphorbia tirucalli* is a medicine plant possessing many bioactive properties. This paper focused on phytochemical screening (alkaloid, flavonoid, saponin, tannin, and anthraquinone), quantification of polyphenol and flavonoids, and activating evaluation of antioxidants and antimicrobial properties against *Xanthomonas axonopodis* of different extracts from *Euphorbia tirucalli* grown in Binh Thuan, Vietnam. The best activity fraction was used for purification and determining bioactive ingredients. The results showed that the phytochemical study revealed the presence of alkaloids, flavonoids, tannins, and terpenoids in the ethyl acetate fraction. Saponin and anthraquinone did not present in all extracts. The content of polyphenol and flavonoid of *Euphorbia tirucalli* stem was in the range of 16.65–106.32 mg EqAG/g and 97.97–450.83 μg QE/g. The ethyl acetate fraction showed higher amounts of polyphenol and flavonoids and antimicrobial activity against *X. axonopodis* than other fractions. The antioxidant (SC50) activity of *Euphorbia tirucalli* stem was in the range of 12.91 ± 0.70 and 528.33 ± 25.15 μg/mL. At concentrations of 5.0 and 7.5 mg/mL, the diameter of inhibition of the ethyl acetate fraction was 14.33 ± 0.76 mm and 17.87 ± 0.57 mm, respectively. The MIC (minimum inhibitory concentration) was 0.156 mg/mL. Scopoletin, gallic acid, and piperic acid got MICs corresponding to 78, 312, and 312 μg/mL, respectively. Scopoletin, gallic acid, and piperic acid were found in the ethyl acetate fraction of Euphorbia tirucalli and exhibited the treatment of citrus bacteria canker and plant diseases.

## 1. Introduction

*Euphorbia tirucalli* L. belongs to the *Euphorbiaceae* family and is a commonly popular herb in traditional herbal medicine. *E. tirucalli* is universally known as aveloz. It is a native of Africa and America. Morphologically, this plant can be described as presenting extremely cylindrical green and branched stem with small leaves that appear only at the beginning of its development, small female and male terminal flowers, and encapsulated fruit containing three ovoid seeds [1]. Various studies indicated that *E. tirucalli* is a valuable source of medicinal compounds with active substances such as phytosterols, triterpenes, diterpenes, polyphenols, tannins, et cetera [2,3,4,5]. The alcohol extract from *Euphorbia tirucalli* stem possessed broad-spectrum antimicrobial activity against *Escherichia coli*, *Proteus vulgaris, Salmonella enteritidis, Bacillus subtilis, Staphylococcus aureus, S. epidermidis, Pseudomonas aeruginosa, Klebsiella pneumoniae, Xanthomonas citri, Candida albicans, C. tropicalis, C. glabreta, Aspergillus niger, A. fumigatus, A. flavus* and *Fusarium oxysporum* [6]. Numerous notices showed that flavonoids in this plant effectively inhibited the bacteria growth due to its ability to form a complex with extracellular proteins of the cell wall and disrupt the microbial membrane, while the polyphenols are toxic to micro-organisms. The tannins in this plant are able to inhibit bacteria by inactivate microbial adhesion, enzymes, and cell envelope transport proteins [7].

In Vietnam, located on the Indochinese peninsula, Southeast Asia, the Pacific coast, *E. tirucalli* showed the presence of arjunolic acid, eriodictyol, quercitrin, afzelin, scopoletin, 3,3′,4- trimethylellagic acid, and gallic acid [8,9]. *Euphorbia tirucalli* is used in popular medicine to treat cancer, antibiotics, and antiviral diseases, popularly known as “giao” or “xuong ca” or “san ho xanh” in Vietnam. Currently, there are no reports on antioxidant properties and antimicrobial activity against *Xanthomonas axonopodis* caused canker disease in the lime tree from the extract of *E. tirucalli.*


Citrus bacterial canker (CBC) is a disease that occurs on the plant and causes damage to the economy of many tropical and subtropical countries. The disease occurs in the plants when the plant is under the attack of bacteria *Xanthomonas axonopodis* pv. citri (synonyms: *X. citri* subsp. citri, *X. campestris* pv. citri, *X. citri* pv. citri) [2,10]. The pathogen causes symptoms on leaves, fruits, and twigs of lime plants. Symptoms of citrus canker include round spots that become brown and corky and are sunken in the center with water-soaked margins surrounded by yellow chlorotic halos. The disease results in economic losses in terms of low quality and productivity of lime fruits and the costs for the disease control [11]. 

Chemicals such as synthetic pesticides are a necessary part of agricultural production processes in the process of technological evolution. The use of chemicals is the best strategy for preventing pre- and post-harvest crop losses caused by insect pests and diseases. However, their residual toxicity causes hazards to a human, animal life, the whole environment, and the ecosystem. Therefore, finding organic matter alternative approaches for increasing the management of plant pathogenic microorganisms is the current inevitable trend. The medicinal plant extracts of plant pathogens inhibition do not render problems to the living environment and the world is interested in ways to substitute chemical substances. Numerous studies have presented medicinal plants as a valuable therapeutic aid for various ailments. These natural plant products, known as bio-pesticides, have long been used in the control of microorganisms causing plant diseases [12].

In the current study, the antibacterial activity of *E. tirucalli* stem extracts (*n*-hexane, ethyl acetate, and butanol) against *X. axonopodis* was evaluated in vitro using the agar well diffusion and minimum inhibitory concentration (MIC) method. The antioxidant properties, phytochemistry compositions, polyphenol content, flavonoid content of different extracts from *E. tirucalli* stem were also analyzed and evaluated. Fraction exhibited the best antibacterial activity was purified for determining the purified substances.

## 2. Results

### 2.1. Phytochemical Screening

The concentrated ethanol extract and different fractions (*n*-hexane, ethyl acetate, and butanol) were positive for flavonoids and negative for saponin and anthraquinones. The ethyl acetate fraction and butanol was a positive reaction for alkaloid and tannin. Otherwise, the *n*-hexane fraction was positive for flavonoids and terpenoids (Table 1).

### 2.2. The Content of Polyphenol and Flavonoids

All fractions of *E. tirucalli* stem contained polyphenol and flavonoids to a varying extent (Table 2). Among the three fractions, the ethyl acetate fraction showed the highest content of polyphenol and flavonoids, corresponding to 106.32 ± 0.49 mg of gallic acid equivalent (GAE)/g and 450.83 ± 0.93 μg quercetin equivalent (QE)/g, respectively. The polyphenol and flavonoids content got the lowest value in *n*-hexane, corresponding to 16.65 ± 0.45 mg GAE/g and 97.97 ± 0.79 μg QE/g, respectively.

### 2.3. Antioxidant Activity Basing on DPPH Assay

As evident from Table 3, the radical scavenging activity of the crude extract ethanol and three fractions was the standard increasing with the concentration elevation. At the maximum concentration (50 µg dry matter (DM)/mL), all fractions gave lower inhibition percentages than the Trolox (Sigma). Trolox was used as the control sample. The ethyl acetate fraction had the highest DPPH radical inhibition activity (96.23 ± 0.13%), followed by butanol fraction (23.37 ± 1.01%). Otherwise, the comparison of SC­50 (µg DM/mL), which is the necessary amount of each sample to decline by 50% the decreasing order: Trolox (SC50 = 8.23 µg DM/mL) > ethyl acetate fraction (SC50 = 12.91 µg DM/mL) > butanol fraction (SC50 = 55.56). However, there was not a statistically significant difference in SC50 of ethyl acetate fraction and Trolox (*p* < 0.05).

### 2.4. Antibacterial Test by Well Diffusion Method

Table 4 showed the antibacterial activities of fractions of *E. tirucalli*, expressed in the size of the inhibition zone of bacterial growth and minimum inhibitory concentration (MIC) against *Xanthomonas axonopodis*. At a concentration of 5 mg/mL, all extracts of *Euphorbia tirucalli* exhibited activity against *X. axonopodis*. The diameters of inhibition ranged from 6.10 ± 0.65 to 14.33 ± 0.76 mm. Meanwhile, the highest inhibition diameter (17.67 ± 0.57 mm) was at a concentration of 7.5 mg/mL, found in ethyl acetate fraction. MIC of ethyl acetate fraction was 0.156 mg/mL. The lowest antibacterial activity with an inhibition zone of 6.10 ± 0.65 mm at a concentration of 5.0 mg/mL appeared in *n*-hexane fraction. There was a statistically significant difference in antibacterial activity of all extracts (*p* < 0.005). In the current study, all fractions of *E. tirucalli* showed the low antibacterial activity and was a statistically significant difference (*p* < 0.005) in comparison to the control antibiotic (Streptomycin). At a concentration of 2.5 mg/mL and 1.25 mg/mL, *n*-hexane fraction did not show activity against *X. axonopodis*. Meanwhile, other extracts exhibited antibacterial activity with inhibition zone ranged from 3.70 ± 0.30 to 12.30 ± 0.79 mm.

### 2.5. Isolation of Substances from Ethyl Acetate Fraction

The results of purification and running spectroscopy NMR showed that ethyl acetate fraction contained scopoletin (Figure 1a), gallic acid (Figure 1b), and piperic acid (Figure 1c) (Table 5). All three substances existed in an amorphous form. The gallic acid and piperic acid were white in color, and scopoletin was yellow in color. Table 5 exhibited the structural data of substances 1, 2, and 3 basing on Appendix A.

The MIC of scopoletin, gallic acid, and piperic acid corresponded to 78, 312, and 312 μg/mL, respectively. The MIC of gallic acid and piperic acid against *Xanthomonas axonopodis* was equal and higher than scopoletin. Scopoletin got the lowest MIC and corresponded to 1/4, compared to gallic acid and piperic acid.

## 3. Discussion

As compared to other fractions, the ethyl acetate fraction of *E. tirucalli* possessed more phytochemical constituents. According to Younes et al. (2018), the extraction of phytochemical constituents may depend on the polarity and the molecular weight of the chosen solvent [13]. The current results were similar to a reported by Orlandar et al. (2015) on *E. tirucalli* extracts [14]. However, the material in the present study did not reveal the presence of saponin and anthraquinone, a difference in comparison to the previous studies. This result allows considering the studied specimen as a potential source of new antimicrobial agents since the saponin represents a group with known toxic components.

Polyphenol and flavonoids are the most important secondary metabolites in plants and wildly spread throughout the plant kingdom [15]. It is universally known for its biological activities and health benefits for plants and humans, such as antibacterial and antioxidant activity. Flavonoids exist in plants in different combinations and depend on numerous factors, such as glycosides linkages and sugar moiety. Phytochemical constituents in the extraction depend on the polarity and the molecular weight of the chosen solvent, also its existence in plants, and these factors play a large role in natural substance isolation. The results represented that extract using moderately polar solvent was more effective than low polarity solvents to get higher content of phenolic and flavonoid. Phytochemical screening was similar to *n*-hexane fraction that was only positive for flavonoid. As compared to other fractions, ethyl acetate fraction of the stem possessed the maximization content of polyphenol that attributed to its good solubility, low toxicity, medium polarity, and high extraction capacity. Likewise, ethyl acetate tends to extract a way better than ether and *n*-hexane and is known to be more natural and safer than other chemicals [16]. Overall, the findings indicated that the ethyl acetate extract of stem *E. tirucalli* was rich in flavonoids and polyphenol contents, which could be the main contributor to their antioxidant properties as many studies affirmed that flavonoids and polyphenol offered the highest ability of scavenging activity in medicinal plants [17,18].

The *n*-hexane fraction with a maximum inhibition percentage at a maximum concentration did not reach 50% DPPH radical inhibition; this result may be due to the low antioxidant activity of this fraction.

In the current study, the results clearly stated that ethyl acetate fraction possessing the highest amounts of polyphenol and flavonoids got the best radical scavenging activities and total antioxidant capacities. The positive correlation between polyphenol and the antioxidant activity using DPPH free radicals confirmed the hypothesis of the positive role of these types of compounds. Miceli et al. (2015) have reported similar results by revealing a positive relationship between polyphenol, flavonoids, and DPPH free radical scavenging activity [19]. These results could allot to the presence of phenolic compounds, which play an efficient role as a hydrogen donator, reducing agents, and singlet oxygen quenchers [20,21]. Many plants from the family *Euphorbiaceae* were used in traditional, complementary, and alternative medicine [22]. Melo et al. (2011) analyzed other species *Euphorbiaceae*, such as *Croton blanchetianus* Baill and *Jatropha mollissima* (Pohl) Baill, finding an IC50 of 94.0 and 55.0 μg/mL, respectively [23]. These results confirm that *Euphorbia tirucalli* L. has high antioxidant potential.

According to Sultan et al. (2016), there was a difference in antibacterial activity for different extracts of *E. tirucalli* that depend on the biologically active phytochemical constituents [24]. In the current study, two fractions (ethyl acetate and butanol) contained alkaloids, flavonoids, tannins, and triterpenoids. According to previous studies, alkaloid has been reported able to inhibit the nucleic acid synthesis of bacteria, whereas the tannins able to give toxic to bacteria by increased their hydroxylation proses [15,25]. Polyphenol disturbs the growth of bacteria by inhibition of c-di-AMP that controls various functions in bacteria [7]. The present study is an important step in developing plant-based pesticides which are eco-friendly for the management of the plant pathogenic bacteria and development of a commercial formulation of botanicals.

Based on the results of the chemical shifts in NMR spectroscopy of three purified substances and previous studies, scopoletin, gallic acid, and piperic acid were determined [26,27,28,29,30].

The 1H NMR spectrum of 3 showed the presence of a 1,2,4-trisubstitutedbenzenoid ring (δ_H_ 7.11, d, *J* = 1.5, 6.85, d, *J* = 9.0, 7.00, dd, *J* = 9.0, 1.5), a dioxygenated methylene group (δ_H_ 6.03, s), and four olefinic protons from δ_H_ 6.86–7.31. The 13C NMR spectrum of 3 presented signals of one carbonyl carbon, seven sp^2^ methines, one acetal methylene, and three substituted aromatic carbons. Careful analysis of the spin-spin coupling of four olefinic protons revealed the presence of two consecutive double bonds through C-_7_-C-_10_. Analysis of the J_H-7_/_H-8_ and J_H-9_/_H-10_ (15.5 Hz) indicated the (E) configuration of these double bonds. HMBC spectrum exhibited cross-peaks of both H-_9_ (δ_H_ 7.31) and H-_10_ (δ_H_ 6.86) to carbon carbonyl C-_11_ (δ_C_ 164.4) that defined the linkage of the 11-COOH at C-_10_. Likewise, HMBC correlations of H-_7_ (δ_H_ 6.86) and H-8 (δ_H_ 6.97) to C-_1_ (δ_C_ 131.3) indicated the connection of the side chain at C-_1_. Last, the position of the methylene group was assigned due to its HMBC correlations to both C-_3_ and C-_4_ (Appendix A). NMR data of 3 closely resembled those of piperic acid, so elucidation 3 was as piperic acid (Takao et al., 2015) [31].

Scopoletin, gallic acid, and piperic acid exhibited antibacterial activity of both Gram (–) and Gram (+). For example, scopoletin belongs to coumarin resistant Gram (–) (*Salmonella typhimurium* (MIC of 250 μg/mL [32], *Pseudomonas aeruginosa* ATCC 27853, and Pseudomonas DMSC 37166 (MIC of 0.66 μg/mL [33], *Prevotella intermadia, Porphyromonas gingivalis,* and *Aggregatibacter actinomycetemcomitans* (MIC from 0.25 μg/mL to 1.0 μg/mL and 0.5 μg/mL to 1.0 μg/mL, respectively [34] and Gram (+) (*Actinomyces naeslundii, Actinomyces israelii,* and *Streptococcus mutans*) [34]. Scopoletin caused the change of the bacterial cell wall similar to antibiotic β-lactam basing on the inhibition of the cell wall formation and the deformation of the bacterial morphology [33]. Gallic acid (polyphenol) in plants possessed numerous bioactivities, for example, antibacterial, antivirus, antioxidant, anti-inflammatory [35]. For example, gallic acid inhibited *X. citri sub sp. citri* that caused ulcers on the lemon tree with MIC of 500 μg/mL [36], *Staphilococcus aureus, Salmonella typhimurium* (MIC corresponding to 1250 μg/mL, and 2500 μg/mL, *Escheriachia coli* ATCC25922 (MIC of 2.5 μg/mL), *Salmonella typhimurium* CMCC(B)50115 (MIC of 2.5 μg/mL), *Shigella flexneri* CMCC(B)51572 (MIC of 2.5 μg/mL), and *Pseudomonas aeruginosa* ATCC27853 (MIC of 2.5 μg/mL) [37], *Pseudomonas aeruginosa* (MIC of 500 mg/mL), *E. coli* (MIC of 1500 mg/mL), *Staphilococcus aureus* (1750 mg/mL), and *Listeria monocytogenes* (2000 mg/mL) [38]. For bacterial gram (+), gallic acid inhibited *Staphilococcus aureus* CMCC(B)26003 (MIC of 0.63 μg/mL), *Bacillus cereus* CMCC(B)63301 (MIC of 2.5 μg/mL), *Staphilococcus epidermidis* CMCC(B)26069 (MIC of 0.63 μg/mL), and *Monlilia albican* CMCC(F)98001 (MIC of 5.0 μg/mL). Gallic acid led to a change in the properties of cell membranes of bacterial. For example, permeability, charge, physical and chemical properties, change hydrophobia, reducing the negative charge on the surface, the surface rupture on the membrane surface, or the hole formation in the cell membrane, and the damage of essential intracellular components, killing bacteria [38]. Zarai et al. (2013) reported piperic acid inhibiting bacteria Gram (-): *E. coli, K. pnuemonia,* and *S. enterica* with MICs of 312.5, 625, and 625 μg/mL, respectively; bacteria Gram (+): *S. epidermidis, S. aureus, S. xylosus, B. subtilis,* and *E. faecalis* with the respective MICs of 78.12, 312.5, 156,25, 156,25, and 312.5 μg/mL [39]. This current result is the basis for the development of organic plant protection drugs and new in comparison to previous studies.

## 4. Materials and Methods

### 4.1. Source of the Plant Material

Aerial parts of the *E. tirucalli* was collected from Phan Thiet city, Binh Thuan province–MG (11°06′01″ North latitude–108°08′28″ East longitude) on April 2019.

### 4.2. Preparation of Plant Extracts

The dried powder of aerial parts of the *E. tirucalli* (6.5 kg) was macerated in 96% ethanol (EtOH) (5 × 15 L) for 24 h at the temperature room. The filtrate was collected using filter paper (Whatman No. 1, USA) and concentrated with a rotary evaporator (Heidolph, Germany) at 45 °C under vacuum condition to obtain the extracts (616.81 g). Continuously, the condensation separation was in turn by using *n*-hexane, ethyl acetate, and butanol to afford different fractions, corresponding to 179.14 g, 114.97 g, and 73.41 g, respectively. Keeping all condensations were at 4 °C until further studies.

### 4.3. Standard Bacteria

Obtaining a pure culture of *Xanthomonas axonopodis* (Strain BLKQ1, MT328595.1) that causes citrus canker disease in the lime tree was from the Biotechnology Department, Nong Lam University Ho Chi Minh City Viet Nam.

### 4.4. Phytochemical Screening

Analysis of preliminary phytochemical compositions on different fractions was according to the description of Brain and Turner (1975) [3] and Evans (1996) [4] with slight modification.

#### 4.4.1. Alkaloids

A total of 2.0 g of the crude extract and three fractions were dissolved individually in 20.0 mL of 1% HCl and filtered. The filtrates then added 10% ammonium hydroxide until the filtrates became alkaline. The mixture continuously added 5.0 mL of chloroform for separating the aqueous and organic layers. The organic layer added 2.0 mL of 1% HCl and treated with Wagner’s reagent for the formation of brown/reddish brown precipitate indicates the presence of alkaloids.

#### 4.4.2. Flavonoids

A total of 5.0 g of the crude extract and three fractions were added to 20 mL of 70% ethanol and heated until boiling. The hydro-alcohol extracts were filtered and treated with a few drops of H_2_SO_4_. The formation of orange color indicates the presence of flavonoids.

#### 4.4.3. Saponins

A total of 1.0 g of the crude extract and three fractions were added to 10.0 mL of distilled water and boiled. After cooling, the obtained extracts were vortexed for 15 s. The formation of foam indicates that the presence of saponin.

#### 4.4.4. Tannin

A total of 10.0 g of different fractions were diluted in 100.0 mL of distilled water and heated on a water bath. The mixture was then filtered and added ferric chloride. The formation of a dark green color indicates the presence of tannin.

#### 4.4.5. Anthraquinones

A total of 3.0 g of the crude extract and three fractions were diluted in 5 mL chloroform and shook for 5 min. The mixture was filtered and added an equal volume of 10% NH_3_ and heated. The formation of pink color indicated the presence of anthraquinone.

#### 4.4.6. Terpenoids

A total of 5 mg of different fractions were in turn mixed to 2 mL of chloroform and 3 mL of concentrated H_2_SO_4_. The formation of reddish-brown color in the inner face indicated that the presence of terpenoid.

### 4.5. Quantification of Polyphenol Content

The amount of polyphenol in *E. tirucalli* extracts was determined with Folin–Ciocalteu reagent according to the method of Singleton and Rossi [5] with slight modification using gallic acid as a standard. Briefly, in 200 µL of extract (2 mg/mL) was added 500 µL of 1/10 diluted Folin reagent and 20% Na_2_CO_3_. The mixture was allowed to stand for 30 min with intermittent shaking, and the absorbance was measured at 730 nm using a UV-Vis spectrophotometer (Jenway 6100, Dunmow, Essex, U.K).

### 4.6. Quantification of Flavonoids Content

The amount of flavonoids in *E. tirucalli* extracts was determined using a modified aluminum chloride assay method [40] and diluting all fractions in ethanol to a concentration of 1 mg/mL. 0.5 mL of extract was in turn mixed with 2 mL of distilled water and 150 μL of 5% sodium nitrate for 6 min. 150 μL of 10% aluminum chloride and 2 mL of 1 M sodium hydroxide was continuously added into the mixture and kept at room temperature for 15 min. Solution absorbance was measured at 320 nm (UV-Visible Ultraspec JANEWAY 7305 spectrophotometer, UK) with a quercetin standard and expressing flavonoid content as μg of quercetin equivalents per gram of sample (μg QE/g of the extract).

### 4.7. Chromatographic Separation and Isolation of Constituents

A fraction of ethyl acetate (EA) was applied to a silica gel CC and eluted with a solvent system of *n*-hexane: ethyl acetate: acetone (5: 1: 1 to 1: 1: 1, *v*/*v*/*v*) to afford 18 fractions EA1-18. The fraction EA3 was fractionated by CC with the solvent system of *n*-hexane: chloroform: ethyl acetate: acetone: acetic acid (350: 100: 40: 25: 10, *v*/*v*/*v*/*v*/*v*) to afford 4 fractions EA3.1-4. EA3.2 was run via CC with methanol and afforded three fractions EA3.2.1-3. EA3.2.1 (166.3 mg) was continuously run via CC with the elute of methanol: water (1: 2 to 1: 1, *v*/*v*) to obtain three compounds 1 (20 mg), 2 (4.3 mg) and 3 (4.6 mg) [8].

### 4.8. Structure Elucidation of the Compounds

Bruker Advance III (500 MHz for 1H NMR and 125 MHz for 13C NMR) spectrometer with TMS as internal standard recorded NMR spectra. Chemical shifts were in ppm with reference of acetone-d_6_ at δ_H_ 2.05, d_C_ 206.26 and 29.84, and of dimethylsulfoxide-d6 at δ_H_ 2.50 and δ_C_ 39.52. TLC with the cover layer of silica gel 60 RP–18 F254S (Merck Millipore, Billerica, Massachusetts, USA) were spots and sprayed with 10% H_2_SO_4_ solution followed by heating. Gravity column chromatography was with silica gel 60 (0.040 0.063 mm) (HiMedia, Mumbai, India). NMR temperature was set in 23 °C.

### 4.9. DPPH Free Radical Scavenging Assay

1, 1-diphenyl-2-picrylhydrazyl (DPPH) free radical scavenging assay was according to the protocol described by Blois [8] and adapted by Brand-Williams et al. [41]. Briefly, 0.5 mL of ethanol containing different concentrations (31.25, 62.50, 125, 250 and 500 µg/mL) of each extract (*v*/*v*) was mixed with 1.5 mL of DPPH• ethanol solution (6 × 10^−5^ M). DPPH• and Trolox solution served as a control and positive control, respectively. All mixtures were kept at room temperature for 90 min and measured the absorbance at 517 nm. Antioxidant activity, expressed as a percentage of DPPH radical scavenging activity (RSA) and SC50, was calculated as follows:% RSA = [(Abs blank − Abs sample)/Abs blank] × 100%(1)
where:

Abs blank: the absorbance of the blank sample;

Abs sample: the absorbance of the test sample

SC50 was the sample concentration requiring to scavenge 50% of DPPH radicals).

All tests were in triplication and expressed under the average and standard deviation.

### 4.10. Antibacterial Activity Assay

Antibacterial activity determination of different fractions such as *n*-hexane, ethyl acetate, and butanol from stem *E. tirucalli* was by using the agar well diffusion method [42] with modification. Each of extract *E. tirucalli* (1.25, 2.5, 5 and 7.5 mg DM) was prepared by addition of dimethyl sulfoxide (DMSO) solvent 10% [7]. The isolated bacteria were incubated into 10 mL sterile nutrient broth (NB) and kept overnight at 37 °C for centrifuging at 1000 rpm for 15 min. The supernatant in sterile distilled water and adjusting of the concentration to 0.5 McFarland Standard with sterile NB. The agar medium was punched with a diameter of 5.0 mm and filled 60 µL of extracts. The positive and negative control were commercial Streptomycin (0.01 mg/mL concentration) and 10% of solvent DMSO, respectively. These plates were allowed to stand for 5 min for the diffusion of extract to take place. The plates were then kept at 37 °C for 48 h and measuring the zones of inhibition (clear zone around each well) in millimeter (mm). Each experiment was in triplication.

### 4.11. MIC Assay

Determination of the minimum inhibitory concentration (MIC) was by using the broth microdilution method [43,44,45,46] in Nutrient agar broth (Himedia). The maximum active stem extract of *E. tirucalli* was dissolved in the solution (water: DMSO, 95: 5) according to the serial double dilutions into a 96-well microtiter plate over the range of 5.0–0.078 mg/mL. The mixture was added 50 µL resazurin (2 mg/mL) and kept at 37 °C for 30 min for the determination of MIC. Resazurin was an indicator of bacteria growth that metabolized and changed it into pink. The bacterial did not grow in the wells that do not change color.

### 4.12. Statistical Analysis

All determinations were conducted in triplicates and expressed as mean ± SD. A significant difference (*p* < 0.05) was analyzed and compared using the Duncan multiple range test in software IBM SPSSS Statistics 20. SC50 is the concentration sufficient to obtain 50% of a maximum effect of 100%.

## 5. Conclusions

In summary, scopoletin, gallic acid, and piperic acid against *Xanthomonas axonopodis* at MIC of 78, 312, and 312 μg/mL, respectively, was isolated from a fraction of ethyl acetate of *Euphorbia tirucalli* grown in Vietnam. Scopoletin, gallic acid, and piperic acid were analyzed and demonstrated basing on 1H-NMR, 13C-NMR, and HMBC. Three substances existed in the ethyl acetate fraction. All fractions from *Euphorbia tirucalli* possessed DPPH free radical scavenging activity and antibacterial activity. Phytochemical (saponin, anthraquinone, terpenoids, alkaloids, flavonoids, and tannin), antioxidants (polyphenol, flavonoids), and antimicrobial existed in Vietnam *Euphorbia tirucalli*. Saponin and anthraquinone did not exist in the initial ethanol extract. Terpenoids occurred more than alkaloids, flavonoids, and tannin. Flavonoid and tannin were a little similar. The plant contained polyphenol and flavonoids that can serve as natural sources of antioxidants and antimicrobial agents. Future study is recommended to further purify and examine individual bioactive compounds from these extracts and evaluate their inhibition activity of *X. axonopodis*.

## Figures and Tables

**Figure 1 molecules-26-00941-f001:**
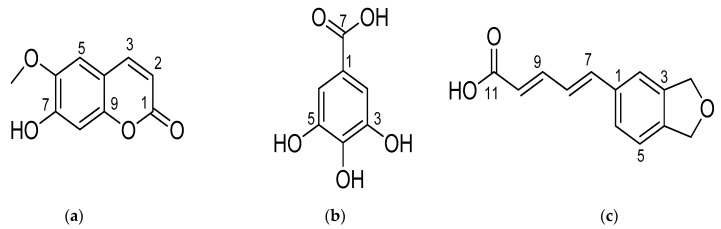
(**a**) Structure of substance 1 (scopoletin); (**b**) structure of substance 2 (gallic acid); (**c**) structure of substance 3 (piperic acid).

**Table 1 molecules-26-00941-t001:** Phytochemistry compositions in different fractions of *E. tirucalli.*

Solvent	Alkaloids	Tannin	Saponin	Flavonoids	Anthraquinon	Terpenoids
Concentrated ethanol extract	++	+	-	++	-	+++
*n*-hexane	-	-	-	+	-	+
Ethyl acetate	++	+	-	++	-	+++
Butanol	+	+	-	+	-	-

^“–”: negative; “+, ++, and +++”: Low, moderately present, highly present.^

**Table 2 molecules-26-00941-t002:** Polyphenol content in different extracts of *E. tirucalli.*

Samples	Polyphenol Content (mg GAE/g)	Flavonoid Content (μg QE/g)
Concentrated ethanol extract	185.28 ± 5.37	824.15 ± 25.55
*n*-hexane	16.65 ± 0.45	97.97 ± 0.79
Ethyl acetate	106.32 ± 0.49	450.83 ± 0.93
Butanol	61.87 ± 0.82	273.34 ± 0.46

**Table 3 molecules-26-00941-t003:** Radical scavenging activity (% RSA) and SC50 of different extracts from stem *E. tirucalli*, determined by DPPH method.

Samples	Concentration% RSA					
	3.125 (µg DM/mL)	6.25 (µg DM/mL)	12.5 (µg DM/mL)	25 (µg DM/mL)	50 (µg DM/mL)	SC50 (µg DM/mL)
Concentrated ethanol extract	10.94 ± 2.40	19.85 ± 1.75	34.95 ± 1.89	46.48 ± 0.61	85.56 ± 0.13	14.93 ± 0.85 ^a^
*n*-hexane	0.53 ± 1.33	0.96 ± 1.33	1.40 ± 1.83	2.52 ± 1.06	4.33 ± 1.92	528.33 ± 25.15 ^c^
Ethyl acetate	13.93 ± 2.40	20.38 ± 1.75	38.11 ± 1.89	77.08 ± 0.61	96.23 ± 0.13	12.91 ± 0.70 ^a^
Butanol	2.93 ± 1.33	5.09 ± 1.31	8.24 ± 1.20	9.32 ± 0.31	23.37 ± 1.01	55.56 ± 0.51 ^b^
The control (Trolox)	16.5 ± 1.06	32.23 ± 2.40	68.53 ± 1.54	132.5 ± 2.48	235 ± 0.15	8.23 ± 0.23 ^a^

^Values in the same row with different superscript letters were significantly different at *p* < 0.05 (mean ± SD, n = 3).^

**Table 4 molecules-26-00941-t004:** Effect of different extracts from stem *E. tirucalli* on bacterial.

Concentration (mg/mL)	Zone of Inhibition (mm)		
	*n*-Hexane	Ethyl Acetate	Butanol
1.25	-	10.57 ± 0.80 ^b^	3.70 ± 0.30 ^b^
2.5	-	12.30 ± 0.79 ^c^	7.63 ± 0.65 ^c^
5.0	6.10 ± 0.65 ^b^	14.33 ± 0.76 ^d^	10.03 ± 0.55 ^d^
7.5	8.67 ± 0.76 ^c^	17.67 ± 0.57 ^e^	11.03 ± 0.35 ^e^
Streptomycin (0.01 mg/mL)	20.13 ± 0.61 ^d^	20.13 ± 0.61 ^f^	20.13 ± 0.61 ^f^
DMSO 10%	-	-	-

^Different superscript letters in the same row exhibited a significant difference (*p* < 0.05) (mean ± SD, n = 3); “-” non-Inhibition.^

**Table 5 molecules-26-00941-t005:** Data of NMR spectroscopy of substance 1, 2, and 3.

Position	Substance 1 (Scopoletin)		Substance 2 (Gallic Acid)		Substance 3 (Piperic Acid)	
	δ_H_, J (Hz)	δ_C_	δ_H_, J (Hz)	δ_C_	δ_H_, J (Hz)	δ_C_
1				122.2		131.4
2			7.14 (1H, s)	110.4	7.11, d, 1.5	108.3
3	6.17, d, 9.5	112.4		146.4		148.1
4	7.84, d, 9.5	143.7		139.6		148.4
5	7.19, s	109.1		146.4	6.85, d, 9.0	105.4
6			7.14 (1H, s)	110.4	7.00, dd, 9.0, 1.5	120.9
7		154.1		170.5	6.86, d, 15.5	137.5
8	6.80, s	102.8			6.97, dd, 15.5, 10.5	122.4
9					7.31, dd, 14.5, 10.5	141.8
10		111.2			6.86, d, 14.5	125.8
11						164.4
3-/5/6-OCH_3_	3.90	55.7				
O-CH_2_-O					6.03	101.4

## Data Availability

Datas are available from the authors.

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
