# Peer review of "Phytochemical Screening and Evaluation of Antioxidant Properties and Antimicrobial Activity against Xanthomonas axonopodis of Euphorbia tirucalli Extracts in Binh Thuan Province, Vietnam"

_molecules, 2021, doi:10.3390/molecules26040941_

Round 1
Reviewer 1 Report
The authors have considered some of the Reviewer comments in the previous submission. However, my main concern is that additional experiments are needed; it is necessary to perform at least a complementary antioxidant experiment (ABTS or another technique) to characterize these extracts.
Author Response
Dear Reviewer 1
Thankful for your help very much
The authors revised the manuscript "molecules-1050914" according to the comment of reviewer 1
Specifically, line 132 and 133, MIC of scopoletin, piperic acid, and gallic acid were added.
Best regards!
Reviewer 2 Report
The manuscript has been revised very little.
I found some items that need to be supplemented or corrected.
- In the description of the phytochemical screening, the terms little, normal and much were used. However, there is no information as to what exactly these terms mean.
- Why was the ethanol extract not used to determine the antioxidant and antimicrobial activity. The results presented in Table 3 show that this extract had the highest content of the desired compounds.
- New point 2.5: Description needs to be corrected. The first sentence suggests that five compounds were obtained, including three named scopoletin (the colon is missing). Why is there a reference to Table 6 in the phrase "Scopoletin was yellow in color"?
- NMR spectra should be included in Supplementary Information, not in the text of the manuscript. (it would be better: Figure 1. 1H NMR (type of solvent) spectrum of …(name of compound)).
- Figures 7 and 9 are scans, which is unacceptable. The structures of these three compounds should be listed as one figure, not three.
- In addition, the consecutive carbons in compounds 1 and 2 should be numbered, similar to the compound 3.
- Compound 2 - 13C NMR spectrum is missing (1H NMR is instead of it).
- Figure 6 is not the 13C spectrum.
- The chemical shifts of the individual signals in Table 6 for compound 2 do not correspond to the shifts of these signals seen in the 1H NMR spectrum.
- Point 4.7 - on what basis were subsequent fractions selected for the separation?
- Where are the results obtained from HR-ESI-MS, mentioned in point 4.8?
- Where are the results of MIC assay mentioned In point 4.11?
Minor remarks:
- What do the three dots in line 47 mean?
- Line 88, Table 3 – ethnaol or ethanol?
- Table 1 – flavanoids, Line 95 - flavonoids, Table 3 – flavonoid. So how should it be?
Author Response
Dear Reviewer 2
Thankful for your help very much
The authors revised the manuscript "molecules-1050914" according to the comment of reviewer 2
Specifically,
- The terms little, normal and much were replied by using low, moderately present, and high present, respectively. (line 93)
- The paper focuses on the evaluation of bioactive and finding the substances, so the authors only present on various extracts from crude ethanol.
- New point 2.5 and citation of table 6 were adjusted.
- NMR spectra should be removed into Supplementary Information.
- Figures 7 and 9 were revised and added the HMBC spectrum. Specifically, figure 7 became to HMBC spectrum, the old figure 7, 8, and 9 become to figure 8.
- The compounds 1 and 2 were numbered, similar to the compound 3.
- Compound 2 - 13C NMR spectrum is replied.
- Figure 6 is the 13C spectrum belonging to inadequate.
- The chemical shifts of the individual signals in Table 6 for compound 2 is revised.
- Point 4.7 - added the reference.
- HR-ESI-MS, mentioned in point 4.8, is removed.
- MIC assay mentioned In point 4.11, is added in line 132 and 133.
Minor remarks:
- three dots in line 47 mean, was revised.
- Line 88 and Table 3 were revised to ethanol
- Flavonoids were used uniform in the manuscript.
Thank you very much
Best regards!
Reviewer 3 Report
Where is Table 2?
Figures 1 and 2 are identical. Fig 2 not 13C-NMR spectrum.
In line 88 and table 1 hexane is still not used in the correct form.
In line 133: (Some substances isolating from fraction ethyl acetate) it is not correct form, all text needs to be corrected and correct sentences written in English.
Gallic acid and piperic acid were white in color. Scopoletin was yellow in color (Table 6). What does this sentence mean?
7; 8; 9 images are very low quality. One lighter, the other darker.
Again used in line 187 DDPH, although two lines above 185 you have corrected DDPH for DPPH.
In line 238 it is not correct using at the temperature room.
In line 240 crude condensation, you think crude extracts?
4.7 section does not provide correct information. I think you would need HPLC analysis because how can you using the CC method to confirm that there are 18 compounds?
Author Response
Dear Reviewer 3
Thankful for your help very much
The authors revised the manuscript "molecules-1050914" according to the comment of reviewer 3
Specifically,
Table 2 has been edited for the correct order
Fig 2 is revised.
n-hexane was used uniform in the manuscript.
In line 133: was revised.
Citation of table 6 in the manuscript was revised.
7; 8; and 9 images quality were improved.
DPPH was used uniform in the manuscript
In line 238, was revised.
In line 240, was revised to crude extracts.
4.7 section. after running the silica gel CC, 18 fractions were collected, not compounds.
Thank you very much
The authors hope to your help
Best regards!
Round 2
Reviewer 1 Report
The additional antioxidant experiment has not been performed, but in general, the manuscript has been improved.
Only small details:
-In the abstract, please describe MIC (minimum inhibitory concentration)
-Table 5 must be edited
-Section 4.8. Please indicate the temperature of the NMR spectra recording.
-Please indicate that chemical shift assignments of the 1H NMR spectra were based on the literature.
-Conclusion section needs to be improved. Please state the most important outcome of your work. Do not summarize the points already presented in the manuscript.
Author Response
Dear
The reviewer
The authors revised the manuscript according to the comment
Specifically,
-In the abstract, added (minimum inhibitory concentration)
-Table 5 was edited
-Section 4.8. the temperature of the NMR spectra was recorded.
-The chemical shift assignments of the 1H NMR spectra were based on the literature, added.
-Conclusion section was improved.
Thankful for your help
Best wish for you
Best regards!

Reviewer 2 Report
The authors have partially corrected the manuscript, but I still have a few comments.
- The results obtained from MIC assays were added in lines 132-133, without any commentary. It would have been better if the Authors had included these data as an additional section 1.6 (after the information about the isolated compounds). It would be helpful to add at least one sentence of commentary there.
- In section 2.5 in line 136, instead of scopoletin (Fig. S1.), gallic acid (Fig. S2.), and piperic acid (Fig. S3.) it should be scopoletin (Fig. 1a), gallic acid (Fig. 1b), and piperic acid (Fig. 1c). In line 139 the fragment „NMR spectroscopy of substances 1, 2, and 3” is unnecessary. In line 142, instead of Figure 8, it should be Figure 1.
- Table 5 still has errors:
Substance 1 - signals from carbons C-2 (160.4) and C-9 (150.9) are not present on the NMR spectrum
Substance 3 - the signal coming from carbon C-1 (132.5) is not present on the NMR spectrum
The entries in column 1 are: 4-OH, 5-OH, 6-OH. For what purpose?
4 In the manuscript the sequence of the compounds is as follows: scopoletin, gallic acid, piperic acid. Whereas in SI it is different: gallic acid, scopoletin, piperic acid. This should be standardized.
- The newly added fragment in lines 195-208 is not very clear with regard to the discussed compound, especially fragments:
a deoxygenated methylene group (dH 7.11, s)
an aldehyde group (dH 9.65, d, 8.0),
one hydroxyl group (dH 7.95),
one methoxy group (dH 3.91)
two aromatic protons (dH 7.09 x 2),
a (E)-configured double bonds (dH 7.57 d, 15.5, 6.70, dd, 15.5, 8.0),
There is no aldehyde, hydroxyl, or methoxyl group in compound 3, but there is a carboxyl group. What is a deoxygenated methylene group? Most of the signals mentioned here are not in the spectrum description. Moreover, what is dH?
Author Response
Dear
The reviewer
The authors revised the manuscript according to the comments
Specifically,
- The results obtained from MIC assays were discussed (Line 144-147).
- In section 2.5 in line 136, Fig. S1., Fig. S2., and Fig. S3. were replied by Fig. 1a, Fig. 1b, and Fig. 1c, respectively. In line 139 the fragment, NMR spectroscopy of substances 1, 2, and 3” is removed. In line 142, Figure 8 was replied by Figure 1.
- Table 5 was revised
4-OH, 5-OH, and 6-OH in column 1 were removed.
4. In the manuscript and SI, the order of scopoletin, gallic acid, and piperic acid. was standardized.
- The newly added fragment in lines 195-208, dH was adjusted by δH
There is no aldehyde, hydroxyl, or methoxyl group in compound 3, but there is a carboxyl group. What is a deoxygenated methylene group? Most of the signals mentioned here are not in the spectrum description. were revised.
Thankful for your help
Best wish for you
Best regards!

Reviewer 3 Report
Thanks for the corrections.
Author Response
Dear
The reviewer
Thankful for your help
Best wish to you
Best regards!
This manuscript is a resubmission of an earlier submission. The following is a list of the peer review reports and author responses from that submission.
Round 1
Reviewer 1 Report
The authors reported article: Phytochemical Screening and Evaluation of Antioxidant Properties and Antimcrobial Activity Against Xanthomonas Axonopodis of Euphorbia Tirucalli Extracts in Binh Thuan Province, Vietnam.
The chosen topic is very relevant to obtain biologically active extracts from Xanthomonas Axonopodis of Euphorbia and to study their chemical composition and antioxidant activity. In the introduction, the authors have briefly described the topicality of the topic and set the goal to obtain extracts using solvents of different polarity and to characterize the obtained extracts.
Characterization of the isolated extracts, such as polyphenol and flavanoid content and antioxidant activity, however, the amount of data obtained is insufficient, additional analyzes such as chromatographic analysis would be required to confirm and supplement the existing results. It is not clear and described how the extracts are obtained, partly described as the ethanol extract is obtained, but is not understood as in line 187 - at the extraction is written and then for 24 at the temperature room. What is meant by 24? Hours or another unit of measure? In line 183 it would be more correct to write collected, e.t.c.
An extensive revision of English is needed, because in this style of English it is not even clear at times what the authors have written.
It is also worth mentioning the bibliography, for example, the title of 19 literature sources is not given in full, but in part. Also, the work must have a uniform style, as in the 223 line ml, but in elsewhere of the work mL. Lines 163 and 165 have an incorrect abbreviation for the antioxidant activity method DPPH instead of DDPH.
Reviewer 2 Report
The manuscript describing the use of Euphorbia ticuralli extracts as an antibacterial agent is vague and carelessly written. The manuscript looks more like a draft. Phytochemical screening is sketchy. Only one method was used to determine the antioxidant activity. The results of research on antioxidant and antibacterial activity are not very encouraging. There is also a lack of information as to which specific compounds can be responsible for the activity of the tested extracts.
Reviewer 3 Report
The manuscript ID: molecules-988433 presents the phytochemical screening and evaluation of antioxidant properties and antimicrobial activity against Xanthomonas Axonopodis of Euphorbia Tirucalli Extracts in Binh Thuan Province, Vietnam.
In general, the presentation and the discussion of the data should be improved. There is a lack of rigor in most of the experiments in this study. The figure and Tables must be improved.
The following weaknesses were noted:
Table 1 provides very general information: ‘positive or negative’ for some phytochemicals in different fractions of E. tirucalli. It is necessary to give each phytochemical content.
n-hexan, n-hexane, hexane?
Why only DPPH test was used to determine antioxidant activity?
In Table 4 (Effect of different extracts from stem E. tirucalli on bacterial) concentration (mg/mL) is in extract dry weight per mL?
Table 4 and Fig 1. Need to be separated; in the present form, they are confusing.
The conclusion section must be improved; it is too general; it is necessary to include a synthesis of key points.
Reviewer 4 Report
Language could be improved.
References should be updated.
Experiments are very preliminary to a well-studied plant.